# Study of the Off-Axis Fresnel Zone Plate of a Microscopic Tomographic Aberration

**DOI:** 10.3390/s22031113

**Published:** 2022-02-01

**Authors:** Lin Yang, Zhenyu Ma, Siqi Liu, Qingbin Jiao, Jiahang Zhang, Wei Zhang, Jian Pei, Hui Li, Yuhang Li, Yubo Zou, Yuxing Xu, Xin Tan

**Affiliations:** 1Changchun Institute of Optics, Fine Mechanics and Physics, Chinese Academy of Sciences, Beijing 100049, China; yanglinciomp@163.com (L.Y.); emozhiai123@163.com (Z.M.); remiellsq@163.com (S.L.); voynichjqb@163.com (Q.J.); zhangjh0620@163.com (J.Z.); zhangv_ciomp@163.com (W.Z.); peijian980102@163.com (J.P.); lh19990581721@163.com (H.L.); lyhang3111@163.com (Y.L.); z_ouyubo@163.com (Y.Z.); bryant23@126.com (Y.X.); 2University of Chinese Academy of Sciences, Beijing 100049, China; 3Center of Materials Science and Optoelectronics Engineering, Chinese Academy of Sciences, Beijing 100049, China

**Keywords:** microtomography, off-axis Fresnel zone plate, eliminating aberration, Seidel coefficient

## Abstract

A tomographic microscopy system can achieve instantaneous three-dimensional imaging, and this type of microscopy system has been widely used in the study of biological samples; however, existing chromatographic microscopes based on off-axis Fresnel zone plates have degraded image quality due to geometric aberrations such as spherical aberration, coma aberration, and image scattering. This issue hinders the further development of chromatographic microscopy systems. In this paper, we propose a method for the design of an off-axis Fresnel zone plate with the elimination of aberrations based on double exposure point holographic surface interference. The aberration coefficient model of the optical path function was used to solve the optimal recording parameters, and the principle of the aberration elimination tomography microscopic optical path was verified. The simulation and experimental verification were carried out utilizing a Seidel coefficient, average gradient, and signal-to-noise ratio. First, the aberration coefficient model of the optical path function was used to solve the optimal recording parameters. Then, the laminar mi-coroscopy optical system was constructed for the verification of the principle. Finally, the simulation calculation results and the experimental results were verified by comparing the Seidel coefficient, average gradient, and signal-to-noise ratio of the microscopic optical system before and after the aberration elimination. The results show that for the diffractive light at the orders 0 and ±1, the spherical aberration W040 decreases by 62–70%, the coma aberration W131 decreases by 96–98%, the image dispersion W222 decreases by 71–82%, and the field curvature W220 decreases by 96–96%, the average gradient increases by 2.8%, and the signal-to-noise ratio increases by 18%.

## 1. Introduction

The fast and sensitive acquisition of 3D data is one of the main challenges in modern biological microscopy. Most imaging methods, such as laser scanning confocal techniques, achieve optical laminarization by changing the planes of confocalization, which cannot be imaged at the same time because the focal planes are not recorded simultaneously [1]. In 1999, Blanchard et al. first proposed a new type of variable spacing conic diffractive element-off-axis Fresnel zone plate [2] that could be widely used in multiplanar imaging, fluid velocity measurement, particle tracking, and other fields [3]. In 2010, Dalgamo et al. first introduced off-axis Fresnel zone plates into a microscopic imaging system for the multiplanar imaging of cells [4]. The off-axis Fresnel zone plate-based tomographic microscopic imaging system reduced the problems of sample light bleaching and light damage caused by strong incident light, and it was suitable for live-cell imaging because of its real-time image capture, which avoided the problem of error image acquisition at different times caused by excessive cell swimming.

In 2011, Feng Y et al. observed two swimming sperms using a tomographic microscopy system based on off-axis Fresnel zone plates, but there were chromatic aberrations and geometric aberrations in the system that affected the imaging quality of the tomographic microscopy system [5]. To solve the chromatic aberration problem in the system, scholars around the world have carried out a series of studies. Feng Y et al. used a pre-dispersion element to correct chromatic aberration and applied it to a tomographic microscopic imaging system to solve the problem of chromatic aberration affecting imaging quality in 2013 [6]. In 2016, S. Abrahamsson et al. used multiple shining gratings and a multi-faceted prism to compensate for dispersion from the shaft Fresnel [7]. In 2018, Kuan He et al. used an automatic algorithm for three-dimensional reconstruction and then corrected the color difference generated in the system [8]. The analysis of the other aspects of a microscopic system has also achieved significant progress. Yuan Xiangzheng et al. combined polarization multiplexing with the deposition of Fresnel wave tabs, achieving unequal spacing micro-imaging in 2019 [9]. In 2020, Lauren W et al. expanded the imaging field of a tomographic microstructure and the cells that could not enter the field of view by designing the diffraction level [10]. The above research work created a more advantageous microscopic system based on a chromatographic microbial microstructure.

Despite recent progress, there are still aberrations in microscopic tomographic systems that affect their imaging quality. In 2012, S. Abrahamsson et al. reduced the spherical aberration in a system by correcting the out-of-focus phase error in the object plane corresponding to the non-zero diffraction orders; however, other geometric aberrations still exist in this type of system [11]. In 2014, H. Liu et al. extracted the background of the original image obtained using a binary mask and fit it using the least-squares method. The final corrected image was obtained by subtracting this aberration polynomial from the original image [12]. This method corrected the aberration with post image processing. To reduce the influence of aberration in a chromatographic microscopy system, the design of the off-axis Fresnel zone plate, the core element of the chromatographic microscopy system, is proposed in this paper based on the calculation of the aberration coefficient of the optical path function according to the Fermat principle. This design allows for both chromatographic microscopy and systematic aberration reduction. The aberrated off-axis Fresnel zone plates are fabricated with wet etching. After the performance test, the tomographic microscopic optical path is set up for cell image acquisition, and the image indexes such as the signal-to-noise ratio and average gradient are used for quantitative analysis.

## 2. Design of Fresnel Zone Plate for Aberration Correction

The ability of off-axis Fresnel zone plates to correct for aberrations is achieved by varying the grating inscription distribution. Fresnel zone plates with different inscription densities have different abilities to correct phase aberrations; therefore, in this research, a laminar microscopy system based on an off-axis Fresnel zone plate was first simulated in ZEMAX software to determine the types of aberrations. The results of the ZEMAX simulation show that when the incident light is 530 nm and the chromatography depth is 1.5 μm, there is spherical aberration, coma aberration, astigmatism, field curvature, and distortion in each diffraction order of the chromatography microscope, and the ±1 diffractive light forms a diffuse spot with a radius of 80 μm on the detector.

### 2.1. Grid Distribution Design and Simulation Verification

In this research, the optical path difference at any point on the substrate was calculated based on the theory of spherical wave geometry by applying Fermat’s principle to the optical range function and performing a series expansion on the function. The final inscribed density function of the aberrated off-axis Fresnel zone plate was obtained [13]. The principle is shown in Figure 1, and the specific mathematical equations are:
(1)$$n_{20}=\frac{\cos^{2}\gamma }{r_{C}}-\frac{\cos^{2}\delta }{r_{D}},$$
(2)$$n_{30}=\frac{\sin\gamma \cdot \cos^{2}\gamma }{r_{C}^{2}}-\frac{\sin\delta \cdot \cos^{2}\delta }{r_{D}^{2}},$$
(3)$$n_{40}=\frac{4\sin^{2}\gamma \cdot \cos^{2}\gamma }{r_{C}^{3}}-\frac{4\sin^{2}\delta \cdot \cos^{2}\delta }{r_{D}^{3}}-\frac{\cos^{4}\gamma }{r_{C}^{3}}+\frac{\cos^{4}\delta }{r_{D}^{3}},$$

The coefficients *n*_20_, *n*_30_, and *n*_40_ are functions of the spherical wave recording parameters (*γ*, *rC*, *δ*, *rD*), *n*_20_ is the defocus of the system, *n*_30_ is the coma of the system, and *n*_40_ is the spherical aberration. The central period of the aberrated off-axis Fresnel zone plate determines the diffraction angle of each order, which further determines the spacing of each diffraction order [4]. According to the size of the existing CCD (charge-coupled device) and the grating equation, the calculation results show that the center period is *d*_0_ = 56 μm and *δ* = 1.47 rad.

According to the mathematical model of Itou [14], the aberration *F*_ij_ can be expressed as:(4)$$F_{20}=\frac{\cos^{2}\alpha }{r_{A}}+\frac{\cos^{2}\beta }{r_{B}}+m\frac{\lambda }{\lambda_{0}}n_{20},$$
(5)$$F_{30}=\frac{\sin\alpha \cos^{2}\alpha }{r_{A}}+\frac{\sin\beta \cos^{2}\beta }{r_{B}}+m\frac{\lambda }{\lambda_{0}}n_{30},$$
(6)$$F_{40}=\frac{4\sin^{2}\alpha \cos^{2}\alpha -\cos^{4}\alpha }{r_{A}{}^{3}}+\frac{4\sin^{2}\beta \cos^{2}\beta -\cos^{4}\beta }{r_{B}{}^{3}}+m\frac{\lambda }{\lambda_{0}}n_{40},$$

Based on the above analysis, the aberration *F_ij_* is determined by both the usage parameters and the recording parameters. To use a specific wavelength and the deter-mined usage parameters, the recording parameters can be chosen reasonably so that a specific *F_ij_* = 0 [15]. Based on the entry arm length of 50 mm, *n*_10_ = 18, *λ* = 530 nm, etc., as well as making *F*_20_ = 0, the comet correction equation *F*_30_ = 0 and the spherical aberration correction *F*_40_ = 0, the calculation results show that:*n*_10_ = 18,(7)
*n*_20_ = 0.214,(8)
*n*_30_ = 0.008327,(9)
*n*_40_ = 0.0002499,(10)

The recording parameters are obtained by substituting in Equations (1)–(3), as shown in Table 1; however, due to the existence of other geometric aberrations in the system, the equations cannot be satisfied simultaneously; therefore, the damped least-squares optimization algorithm is used to optimize the image scattering and field curvature in the region near the calculated results of the recording parameters.

The premise of the damped least-squares optimization algorithm is to take successive values of the parameters. The value of the evaluation function is made smaller and smaller until the best evaluation function is found. The specific algorithm workflow is:The evaluation function is established. The evaluation function of the laminar microscopy system is the root mean square error of the wavefront. Additionally, in order to avoid the optimization process, the central period is too large, leading to the imaging distance exceeding the CCD size. The central period of 56 μm of the aberrated off-axis Fresnel zone plate is considered to be a nonlinear constraint.The range of values of the angle recording parameter γ is calculated based on the central inscribed density and grating equation of the aberrated off-axis Fresnel zone plate. The optical path displacement range (rCmin, rCmax, rDmin, rDmax) is limited according to the optical stage dimensions being set as the boundary constraints.After inputting the calculated record parameters and starting the operation, the optimal solution of the record parameters is solved by finding the minimum value of the evaluation function at the next point step-by-step around the initial value.

The actual recording parameters are finally obtained, as shown in Table 1. Table 2 shows the parameters of the aberrated off-axis Fresnel zone plate. The number of grid lines *n* at any point on the aberrated off-axis Fresnel zone plate with the center of the aberrated off-axis Fresnel zone plate as the origin point is shown in Equation (7). Figure 2 represents the overall grid line distribution of the aberrated off-axis Fresnel zone plate. This distribution can be used for the simulation of the laminar microscopy system based on the aberration off-axis Fresnel zone plate in the following, as well as for the subsequent fabrication of the aberration off-axis Fresnel zone plate.
(11)$$n_{p}=\frac{\left(DP-CP)-(DO-CO\right)}{\lambda },$$

The simulated optical system of the aberrated off-axis Fresnel zone plate chromatography microscopy system is shown in Figure 3. In the optical microscopy system, the off-axis Fresnel zone plate is replaced by a holographic surface. By inputting the recording parameters calculated in the previous section, the simulation results show that a diffuse spot with a radius of 5 μm is formed at the image plane when the incident light is 530 nm and the lamination depth is 1.5 μm.

From Figure 3, it can be clearly observed that the rays pass through the double-glued lens and grating in turn, and finally, imaging is performed at the image plane; however, the 3D profile does not objectively represent the imaging performance of the system, so it is necessary to analyze the imaging quality of the system with the help of other graphics. The general image quality judgment indexes of optical systems include point column diagrams, light sector diagrams, and Seidel coefficients. The comparison of the aberrations corresponding to different diffraction levels of the unimproved and aberrated optical systems is shown in Figure 4, and the point column diagram corresponding to the two optical systems is shown in Figure 5.

In Figure 4, the two plots in the light sector of each diffraction order represent the aberration in the tangential plane and the sagittal plane. The horizontal coordinates represent the normalized incident pupil, and the vertical coordinate is the value of the ray’s deviation from the main ray in the image plane. In the diagram, the larger the vertical coordinate of the peak is, the larger the spherical aberration of the system is. The larger the slope of the aberration curve passing through the origin is, the larger the field curvature in the system is. The longitudinal coordinate of the light sector of the laminar microscope after the system improvement decreases from ±500 μm to ±25 μm. This indicates that the spherical aberration is improved due to the counterbalancing of the defocus and spherical aberration. The slope of the aberration curve at the origin is reduced from 0.95 to 0.7, which indicates that the field curvature is also corrected. Furthermore, the position of the image is closer to the ideal image plane with the improved system.

The system imaging quality is generally judged by comparing the radius of the diffuse spot in the point column diagram. The smaller the radius is, the better the imaging performance is. In Figure 5, the improved chromatography microscopy system shows a significant improvement in the diffusion range compared to the unimproved system. The diffuse spot radius of the improved laminar microscopy system is reduced by 80 μm for the ±1st diffraction order and by 0.2 μm for the 0th diffraction order. Thus, the improved off-axis Fresnel zone plate has the best performance of aberration correction.

The Seidel coefficients are an important tool for evaluating the imaging quality of a system. Each of the coefficients represents a different aberration in the imaging system. The smaller the absolute value of the coefficients is, the smaller the corresponding aberrations are. The Seidel coefficients and aberrations of the original 3D microscopy system based on the off-axis Fresnel zone plate and the improved system are shown in Figure 6. The results show that the aberrations in the chromatographic microscopy system are reduced after the optimization of the off-axis Fresnel zone plate according to the design proposed in this paper. The spherical aberration W040, coma aberration W131, astigmatism W222, and field curvature W220 are reduced by 62–70%, 96–98%, 71–82%, and 96–96% for the 0th and the ±1st diffraction light levels. This proves that the grid line distribution satisfies the expected results.

### 2.2. The Groove Structure Design of the Aberrated Off-Axis Fresnel Zone Plate

The etching depth of the off-axis Fresnel zone plate determines the energy of each diffraction order. A 2nd level off-axis Fresnel zone plate is used, and its actual duty cycle is 1:1. In order to make the improved off-axis Fresnel zone plate satisfy the same diffraction efficiency for each order, the diffraction efficiency of the Fresnel zone plate is analyzed as a function of the etching depth using the software PCGrate. The intersection of the efficiency etching depth curves of 0th and ±1st diffraction order is the required etching depth, as shown in Figure 7. The energy of the first three diffraction levels is all 27.67% of the total incident light energy when *n* = 1.46 and the etching depth is 374 nm.

In this paper, the slot type of aberrated off-axis Fresnel zone plate used is a rectangular slot with a variable period; therefore, the effect of etching depth on its diffraction efficiency at different periods is discussed in this paper. It shows that the difference between the diffraction efficiency of 0th order and ±1st order is 0, and the diffraction efficiency does not vary with the period for a certain duty cycle with an etching depth of 374 ± 3 nm in Figure 8. The rectangular slot may not be achieved due to errors in the fabrication process. The difference between the diffraction efficiency of 0th order and ±1st order is 0 and the diffraction efficiency does not vary with the bottom angle of the slot type when the etching depth is 374 ± 3 nm at a certain period, as shown in Figure 8.

## 3. Fabrication, Performance Test, and Simulation of Aberrated Off-Axis Fresnel Zone Plate

In this research, the wet etching process of amorphous material was used for the fabrication of an aberrated off-axis Fresnel zone plate. The main fabrication process includes substrate cleaning, homogenization, exposition, development, and wet etching [16]. First, the quartz substrate was cleaned. The quartz substrate is composed of quartz material (model 7980-0AA made by Corning, Corning, NY, USA), with a diameter of 76.2 mm and a thickness of 1.5 mm. The organic impurities and metal ions on the surface of the quartz substrate were cleaned using SC-1 liquid (H_2_O: H_2_O_2_: NH_4_OH = 5:1:1) and SC-2 liquid (H_2_O:H_2_O_2_: HCl = 6:1:1), respectively. Then, the quartz substrate was cleaned in the order of toluene→acetone→alcohol→water. After cleaning, the substrate was pre-baked at 120 °C for 30 min, and then the quartz substrate was coated with a photoresist using the spin coating method. The photoresist model is BP212-7P, the spin coating speed was 3000 rpm, the coating time was 30 s, and the thickness of the photoresist was 500 nm. After the coating of the photoresist, the quartz substrate was placed in an oven for film hardening at 90 °C for 20 min. Next, the coated photoresist was exposed using UV mask contact exposure with an illumination level of 45 lux and an exposure time of 25 s. Immediately after the exposure, the photoresist was developed with a 3‰ NaOH solution and post-dried. Finally, the quartz was etched with a BHF solution (40% HF:49% NH3F = 1:8). To achieve 374 ± 3 nm grating slot fabrication, the etching time was 374 s because the BHF solution used in the experiment etches the quartz substrate at a rate of 1 nm/s.

After the fabrication of the aberrated off-axis Fresnel zone plate, the system was built after the performance test. The performance parameters of the aberrated off-axis Fresnel zone plate include the center period, slot shape, and diffraction efficiency. The central period and the slot pattern were measured using a Dimension Icon atomic force microscope with a probe placed at the center of the aberrated off-axis Fresnel zone plate to obtain the microstructure and parameters. The diffraction efficiency test of the aberrated off-axis Fresnel zone plate was carried out in two ways. First, the diffraction efficiency was calculated in the software program PCGrate based on the structural parameters acquired with the Dimension Icon microscope. Next, a single-wavelength off-axis Fresnel zone plate diffraction efficiency test optical system was built using a 516.5 nm laser source. The optical system mainly consists of a laser, an aberrated off-axis Fresnel zone plate, and a detector. Nine points (3 × 3) are measured uniformly on the Fresnel zone plate [17]. The ratio of the energy of each diffraction order to the laser energy was calculated.

After passing the performance test of the aberrated off-axis Fresnel zone plate, the laminar microscopy system was built; however, since the 3D profile of the geometrical optics does not visually show the imaging performance of the system, the optical system described above was simulated based on the physical optics of the VirtualLab Fusion software [18]. The simulation based on physical optics was performed by solving Maxwell’s equations during the imaging process of the whole optical system to achieve field tracing; therefore, the simulation can be used to provide a more intuitive imaging result.

Finally, the optical components are selected to construct an adjustable cage system according to the optical system simulation in the software program ZEMAX, as shown in Figure 4. The microscope light source was white light, and a bandpass filter of 20 nm was taken. At the same time, the aberrated off-axis Fresnel zone plate was placed after two achromatic lenses with a focal length of 100 mm. The off-axis Fresnel zone plate and aberrated off-axis Fresnel zone plate were used for image acquisition experiments of the same specimen without adjustment of the instrument operating parameters or the distances of each optical element.

## 4. Result and Discussion

### 4.1. Performance Parameter Tests of Aberrated Off-Axis Fresnel Zone Plate

After the fabrication, the microscopic slot measurement was carried out on the aberrated off-axis Fresnel zone plate using an atomic force microscope. The results of the measurement are shown in Figure 9. The center period was 58 μm, the left edge period was 55 μm, and the right edge period was 40 μm, with a duty cycle of 0.5. The test results were in accordance with the expectation.

In this paper, the diffraction efficiencies of 0th order and ±1st order in the central region of the aberrated off-axis Fresnel zone plate can be calculated using the software PCGrate with the parameters measured in the slot test. The diffraction efficiency result of the fabricated aberrated off-axis Fresnel zone plate is shown in Figure 10. The total diffraction efficiency of the three diffraction orders reaches 80%. It is 82.97% when the wavelength is 516.5 nm.

The diffraction efficiency test optical system of the single wavelength (516.5 nm) aberrated off-axis Fresnel zone plate is shown in Figure 11. The test was carried out by taking points on the aberrated off-axis Fresnel zone plate, as shown in Figure 11, and the test results are shown in Table 3. The average diffraction efficiency of the nine points on the aberrated off-axis Fresnel zone plate is 82.03%, which is in accordance with the expectation. A comparison between the measured and actual values of the diffraction efficiency for different diffraction orders is shown in Figure 12. The results from PCGrate are calculated with the ideal conditions of an aberrated off-axis Fresnel zone plate, which in practice causes errors with the theoretical values due to the influence of the surface roughness.

### 4.2. Imaging Experimental Analysis

The simulation results of the imaging in VirtualLab Fusion are shown in Figure 13. As shown in Figure 13, the images acquired after the improvement based on this method are significantly improved in terms of the accuracy of imaging and uniformity of intensity compared with the three objects.

In terms of the image metrics, the signal-to-noise ratio and the average gradient were used to quantitatively evaluate the images, as shown in Table 4. The signal-to-noise ratio represents the ratio of the image signal to the noise. The average gradient represents the clarity of the image. The larger the average gradient is, the clearer the image is. Compared with the images with the unimproved chromatography microscope, the signal-to-noise ratio of the images acquired by the improved laminar microscope increases by 18%, and the average gradient increases by 3%.

The laminar microscopy optical system was built, as shown in Figure 14. With no adjustment of the instrument operating parameters and the distances of each optical element, the biological cell sections were subjected to image acquisition using the chromatographic microscopic light path and the achromatic chromatographic microscopic light path, respectively. When the magnification of the microscope objective was 5× and 10×, respectively, the acquired biological cell images are shown in Figure 15 and Figure 16. Figure 15a,b and Figure 16a,b show the images acquired by the unmodified chromatographic microscopy system. Figure 15c,d and Figure 16c,d show the images acquired by the achromatic chromatography microscopy system. As can be seen from Figure 15 and Figure 16, the images acquired by the unimproved chromatography microscopy system are blurred and have distortions at the edges. In contrast, the images obtained by the achromatic chromatography microscopy system were clearer and were able to obtain detailed information about the biological cells. Each image index is shown in Table 5. The quantitative analysis was performed using the same image metrics, such as signal-to-noise ratio and average gradient. From Table 5, it can be seen that the signal-to-noise ratio of the aberrated chromatography microscope system was improved by 18.28% on average compared with the off-axis Fresnel zone plate chromatography microscope system. The average gradient is improved by 2.85% on average; therefore, the results show that the imaging performance of the chromatography microscopy system has been improved by the design method of the off-axis Fresnel zone plate proposed in this paper. The feasibility of this method is fully verified by the experiments. Further, the deviation of the actual imaging results from the physical simulation results is mainly due to the fact that the three-layer object information is digital image information (400 × 400-pixel points) and single wavelength imaging during the simulation.

## 5. Conclusions

This research is based on a holographic plane with interference fringes formed by two exposure points that can reduce the aberration of the system. The aberration coefficient calculation method based on the optical range function theory is used to design the aberration-eliminating off-axis Fresnel zone plate. The feasibility of the above method is verified by comparing a software simulation and an experiment. The simulated results show the reductions in the spherical aberration W040 by 62–70%, coma W131 by 96–98%, image dispersion W222 by 71–82%, and field curvature W220 by 96–96% after replacing the off-axis Fresnel zone plate with the aberration eliminating off-axis Fresnel zone plate. This is theoretically feasible. In the experimental aspect, the microscopic optical path was constructed, the cell images were acquired, and the image indexes were evaluated. The comparison of the image metrics shows that the average gradient increases by 2.8% and the signal-to-noise ratio increases by 18%.

In summary, the improved design of the off-axis Fresnel zone plate, which is the core element of the laminar microscope system, improves all types of geometric aberrations. Additionally, no post-processing of the images is required; therefore, the imaging quality of the chromatographic microscopy system is improved using this method.

## Figures and Tables

**Figure 1 sensors-22-01113-f001:**
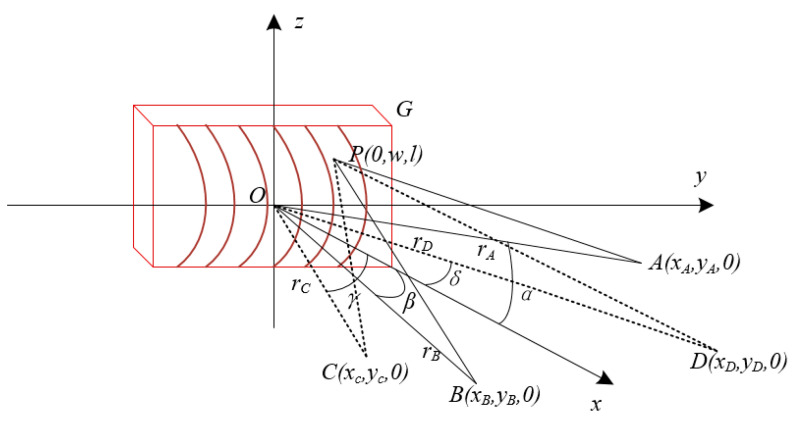
Imaging schematic of aberration axis Fresnel zone plate.

**Figure 2 sensors-22-01113-f002:**
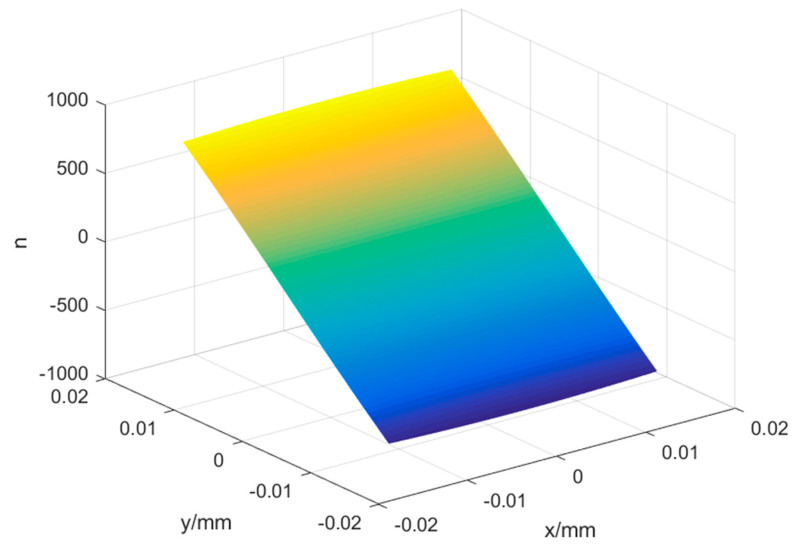
The number of grating lines of the aberrated off-axis Fresnel zone plate.

**Figure 3 sensors-22-01113-f003:**
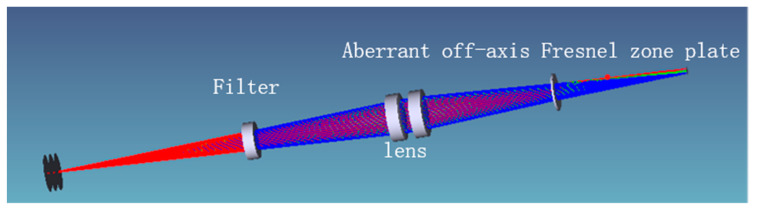
Simulation light path diagram of 3D microscopic imaging system.

**Figure 4 sensors-22-01113-f004:**
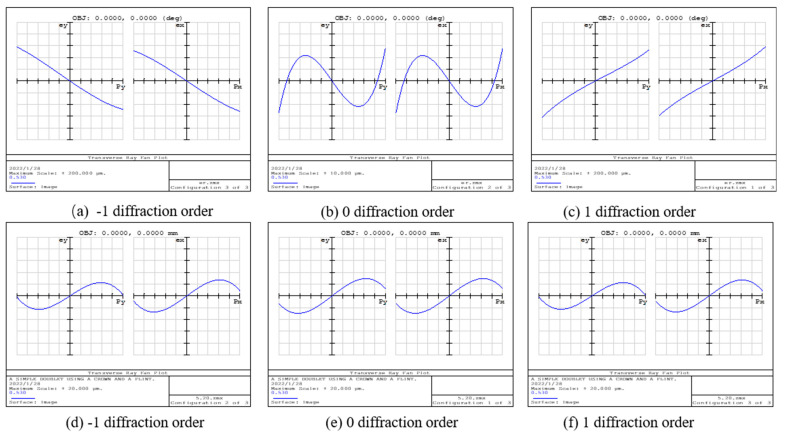
Light sectors calculated by ZEMAX in three orders for the 3D chromatography microscope system before and after aberration correction. (**a**–**c**) Light sectors corresponding to different diffraction levels of unmodified chromatography microscope system. (**d**–**f**) Light sectors corresponding to different diffraction levels of the aberrated chromatography microscope system.

**Figure 5 sensors-22-01113-f005:**
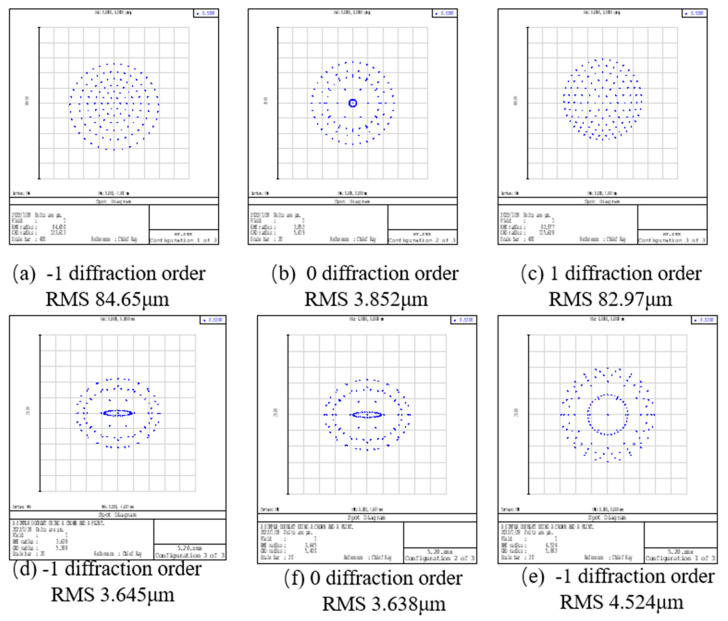
Point column diagram in three orders for the 3D chromatography microscope system before and after aberration correction. (**a**–**c**) Point column diagram corresponding to different diffraction levels of unmodified chromatography microscopy system. (**d**–**f**) Point column diagram corresponding to different diffraction levels of aberrated chromatography microscopy system (compare RMS at the bottom of the diagram).

**Figure 6 sensors-22-01113-f006:**
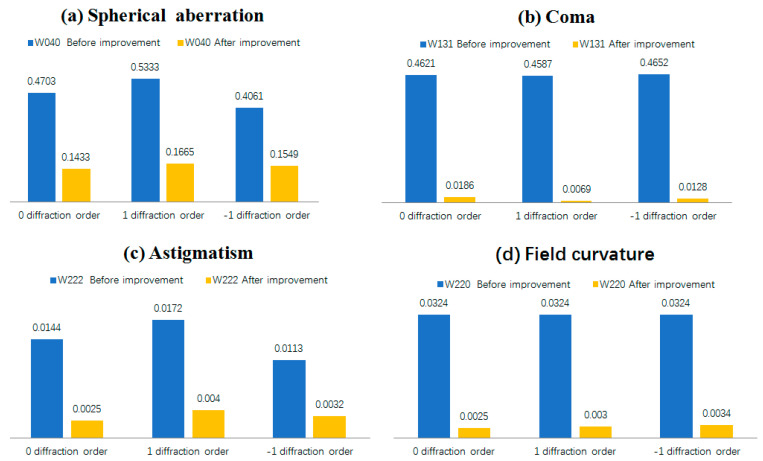
The comparison diagram of the Seidel coefficient of the tomographic microscopic system Blue corresponds to the Seidel coefficient for the unmodified chromatography system. The yellow color corresponds to the Seidel coefficient of the aberrated chromatography system. The adjacent blue and yellow rectangles correspond to one diffraction order of the chromatographic system.

**Figure 7 sensors-22-01113-f007:**
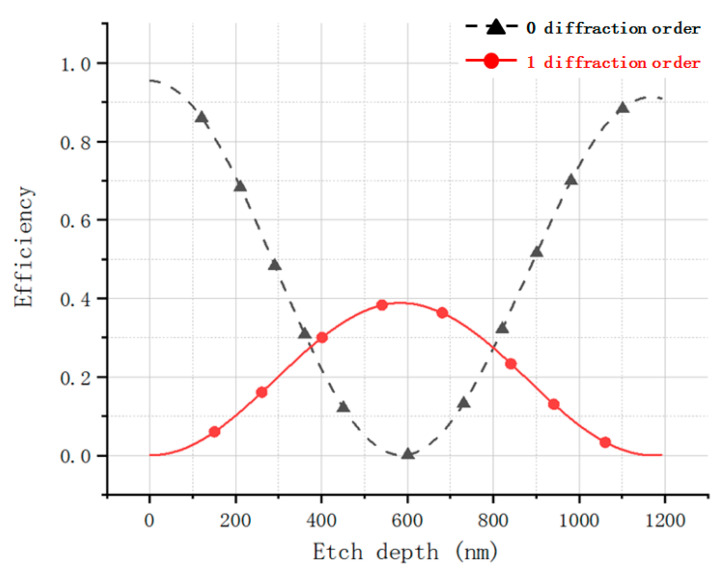
Diffraction efficiency variation curve with etching depth for 0 and ±1 diffraction levels of aberrated Fresnel zone plate (red represents level 0, black represents level ±1).

**Figure 8 sensors-22-01113-f008:**
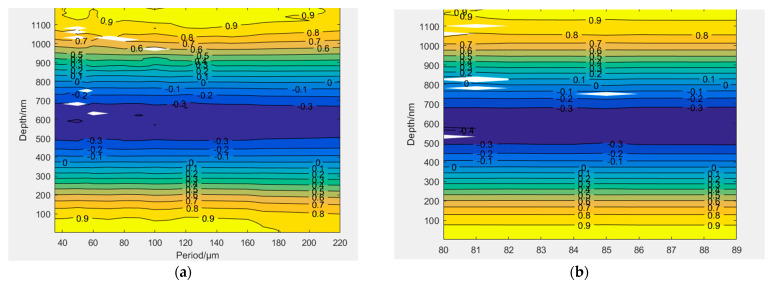
(**a**) Diffraction efficiency difference between level 0 and +1 diffraction versus period (μm) and etching depth (μm) contours (**b**) Diffraction efficiency difference between level 0 and +1 with respect to the contour of the bottom angle (°) and etching depth (μm).

**Figure 9 sensors-22-01113-f009:**
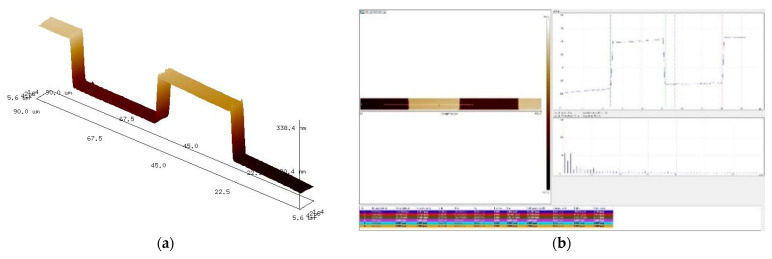
Test results of aberrated off-axis Fresnel zone plate. (**a**) the microstructure (**b**) the atomic force software tests.

**Figure 10 sensors-22-01113-f010:**
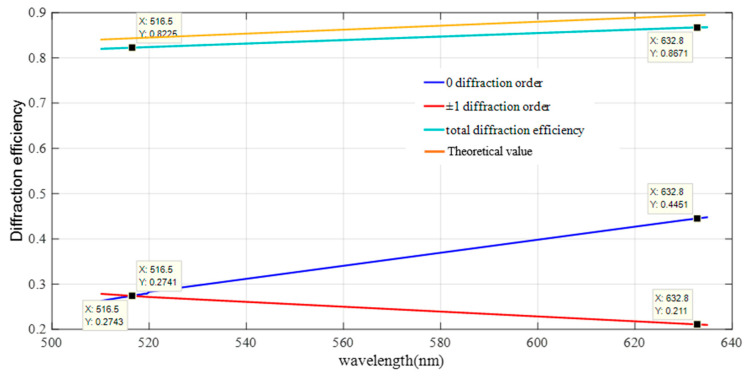
Inversion results of aberrated off-axis Fresnel wave with slice slot type test.

**Figure 11 sensors-22-01113-f011:**
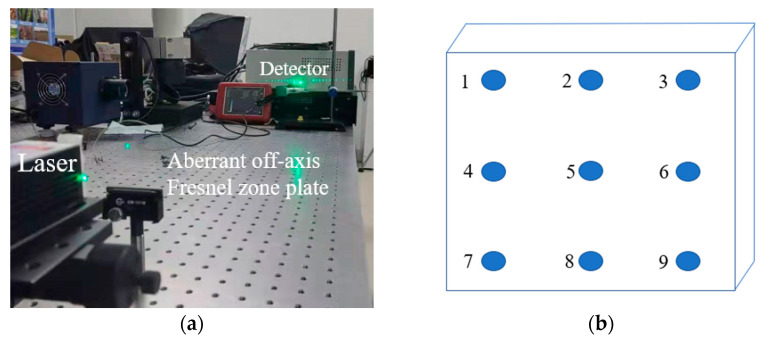
(**a**) Single-wavelength diffraction efficiency test optical path. (**b**) Distribution of sampling points for aberrated off-axis Fresnel waveband diffraction efficiency test.

**Figure 12 sensors-22-01113-f012:**
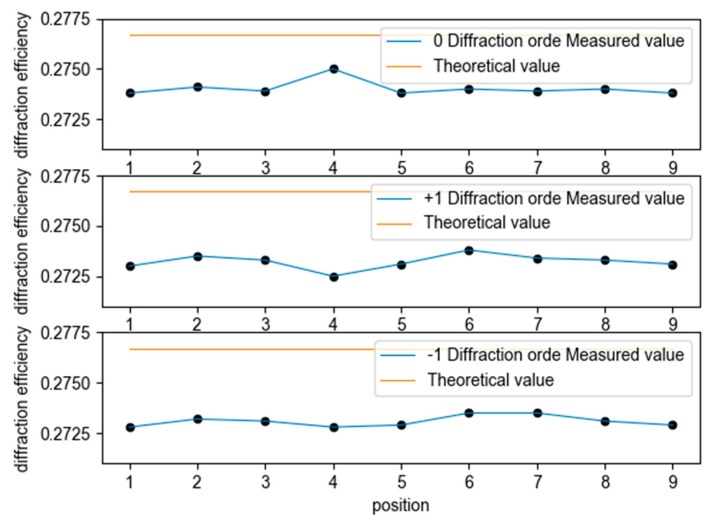
Comparison of theoretical and measured values of diffraction efficiency for different diffraction stages.

**Figure 13 sensors-22-01113-f013:**
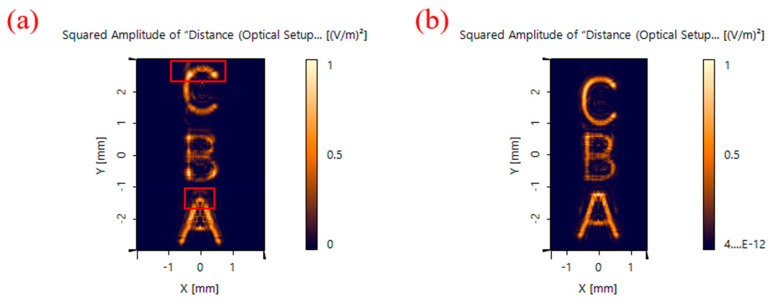
Imaging results: (**a**) Off-axis Fresnel zone plate; (**b**) aberrated off-axis Fresnel zone plate.

**Figure 14 sensors-22-01113-f014:**
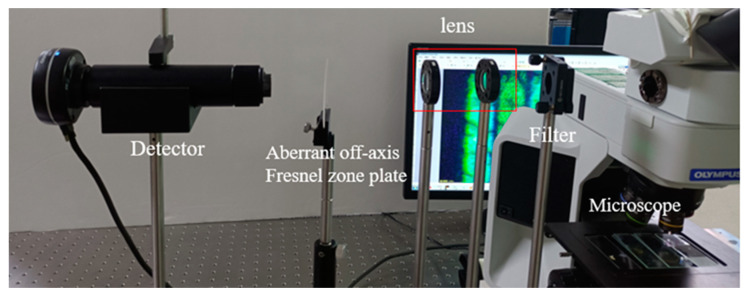
Experimental optical path diagram of chromatography microscope system.

**Figure 15 sensors-22-01113-f015:**
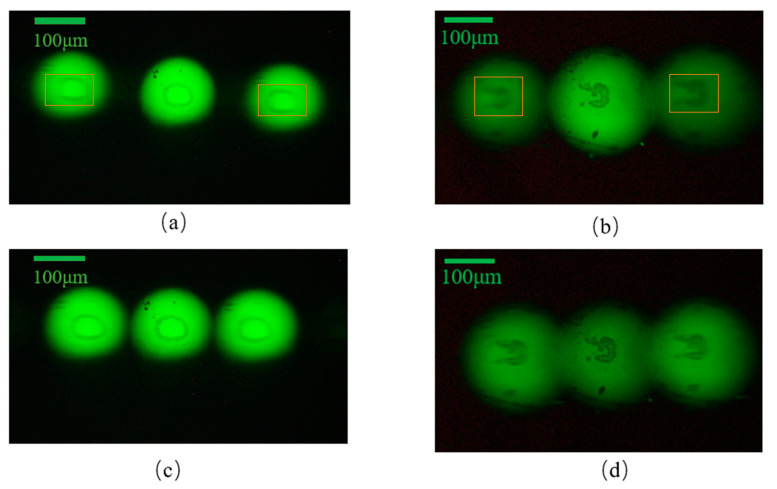
Results of experiment using chromatography microscopy with 5× objective lens with: (**a**,**b**) Off-axis Fresnel zone plate; (**c**,**d**) aberrated off-axis Fresnel zone plate.

**Figure 16 sensors-22-01113-f016:**
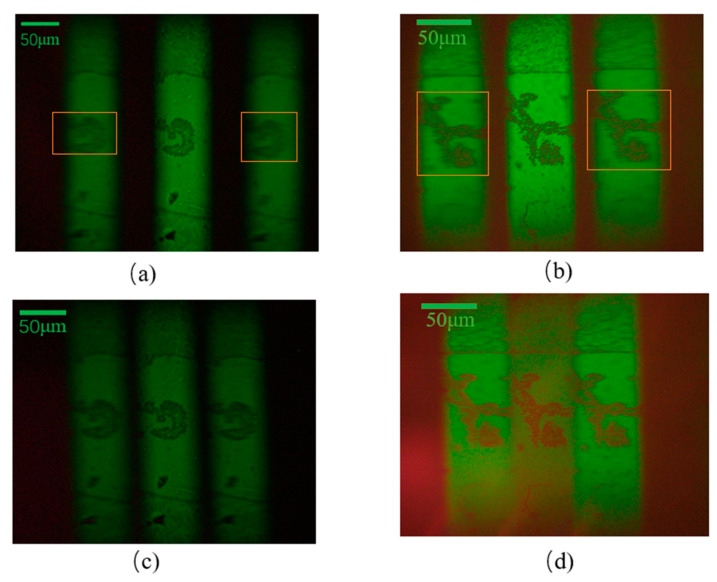
Results of experiment using chromatography microscopy with 10× objective lens with: (**a**,**b**) Off-axis Fresnel zone plate; (**c**,**d**) aberrated off-axis Fresnel zone plate.

**Table 1 sensors-22-01113-t001:** Recording parameters of the aberrated off-axis Fresnel zone plate.

	Spherical Aberration, Coma Aberration	Correction System All Aberrations
*γ*	−1.509 rad	−1.55 rad
*δ*	1.47 rad	1.47 rad
*rC*	59.093 mm	133.9396 mm
*rD*	59.996 mm	133.3251 mm

**Table 2 sensors-22-01113-t002:** The parameters of the aberrated off-axis Fresnel zone plate.

Center Period *d*_0_	Radius *R*	Thickness *b*
56 μm	15 mm	2 mm

**Table 3 sensors-22-01113-t003:** Test results (diffraction efficiency).

No.	0 Level	+1 Level	−1 Level	Total
1	27.38%	27.30%	27.28%	81.96%
2	27.41%	27.35%	27.32%	82.08%
3	27.39%	27.33%	27.31%	82.03%
4	27.50%	27.25%	27.28%	82.03%
5	27.38%	27.31%	27.29%	81.98%
6	27.40%	27.38%	27.35%	82.13%
7	27.39%	27.34%	27.35%	82.08%
8	27.40%	27.33%	27.31%	82.04%
9	27.38%	27.31%	27.29%	81.98%

**Table 4 sensors-22-01113-t004:** Laminar microscopy optical path image index.

Image Index	SNR [dB]	Average Gradient
Off-axis Fresnel zone plate	7.8143	6.0893
Aberrated off-axis Fresnel zone plate	9.3562	6.2719

**Table 5 sensors-22-01113-t005:** Laminar microscopy optical path image index.

Image Index	Off-Axis Fresnel Zone Plate	Aberrated Fresnel Zone Plate
SNR (5×) group 1	17.5556	20.8156
SNR (5×) group 2	21.5643	25.4458
SNR (10×) group 3	21.8550	25.8981
SNR (10×) group 4	21.7625	25.7106
Average gradient (5×) group 1	2.7128	2.7889
Average gradient (5×) group 2	2.7145	2.7905
Average gradient (10×) group 3	3.5359	3.6419
Average gradient (10×) group 4	2.7323	2.8096

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
