# Peer review of "Study of the Off-Axis Fresnel Zone Plate of a Microscopic Tomographic Aberration"

_sensors, 2022, doi:10.3390/s22031113_

Round 1

Reviewer 1 Report

This paper is devoted to three-dimensional microscopic imaging systems improvement. The authors presented a new design of an off-axis Fresnel zone plate which is a core element of the proposed imaging system. The study has scientific novelty and will be useful to researchers. There seem to be no glaring technical errors in the manuscript. Theoretical and experimental studies are in quite good agreement and could reinforce the conclusions. However, there are major revisions I would recommend before the manuscript could be accepted for publication

1) The paper lacks some important references related to the solution of the geometric aberrations problem:

https://doi.org/10.1016/S0304-3991(02)00139-0

https://doi.org/10.1364/AO.53.000748

https://doi.org/10.1016/j.abb.2015.05.010

https://doi.org/10.1038/nmeth.2277

Unfortunately, the authors did not fully compare their methods and results with the recent studies in this area and did not demonstrate the advantages of their approach to improving the imaging system compared to existing techniques in a proper way. However, it could be shown from obtained results.

2) The conclusion should contain quantitative characteristics that confirm the findings. Authors should accent on the advantages of their approach compared to the existing techniques of other researchers.

3) Unfortunately, Figures (mostly 4-9, 15-16) are difficult to parse and interpret. Axes are not visible. The Figure captions do not fully reflect the information presented on plots. The reader has to look for a detailed description in the text. In the presented quality of the figures, it is not possible to confirm the conclusions presented in the text of the manuscript. I believe the improvement of the material presentation in the figures will allow removing this remark.

4) There is a poor description of figures 15-16. What do they demonstrate?

5) There are some stylistic errors (line 166: "...in the diagram" is used twice), typos (line 60: "...et al. Combined..." should be lowercase, line 62: "...et al. By..." should be lowercase, etc.) and extra sentences not related to the work (line 88-92: "Research manuscript... ... prior to publication.") in the manuscript text. It is advisable to proofread the text for such errors.

Author Response

请参阅附件。

Reviewer 2 Report

In this paper, the authors present a design of an off-axis Fresnel zone plate based on double exposure point holographic surface interference aberration elimination is proposed to reduce the effect of aberration. The simulation and experimental verification are carried out employing the Seidel coefficient, average gradient, and signal-to-noise ratio. Which is important for the fast and sensitive acquisition of 3D data. This article is clear, concise, and suitable for the scope of the journal. Several small suggestions are supplied:
1. Suggest avoiding using “new” for the design.
2. Suggest the authors give more detail about Fig.2.
3. Suggest the authors give more detail about Fig.15 and 16.
4. Suggest the authors improve the draft with a native speaker.
